# Chrononutrition: Potential, Challenges, and Application in Managing Obesity

**DOI:** 10.3390/ijms26115116

**Published:** 2025-05-26

**Authors:** Siti Aisyah Fuad, Rehna Paula Ginting, Min-Woo Lee

**Affiliations:** 1Department of Integrated Biomedical Science, Soonchunhyang University, Cheonan 31151, Republic of Korea; siti.aisyah@sch.ac.kr (S.A.F.); rehnapaula@sch.ac.kr (R.P.G.); 2Soonchunhyang Institute of Medi-Bio Sciences (SIMS), Soonchunhyang University, Cheonan 31151, Republic of Korea

**Keywords:** chrononutrition, circadian rhythm, obesity, metabolism, meal-timing

## Abstract

The circadian clock orchestrates nearly every aspect of physiology, aligning metabolic processes with environmental cues, such as light and food intake. While the central pacemaker in the suprachiasmatic nucleus synchronizes peripheral clocks across key metabolic tissue, feeding behavior emerges as the dominant cue for peripheral clock alignment. This interaction reveals a crucial link between circadian biology and metabolism. Disruption of these processes, whether from shift work, irregular eating patterns or lifestyle misalignment, has been strongly associated with metabolic disorders, including obesity, insulin resistance and cardiometabolic diseases. Within the field of chrononutrition, strategies, such as time-restricted feeding (TRF), have gained attention for their potential to restore circadian alignment and improve metabolic health. However, translational gaps persist, as most mechanistic insights are derived from nocturnal murine models, limiting their applicability to diurnal human physiology. Moreover, human studies are confounded by interindividual variability in chronotype, behavioral patterns, and dietary compliance. This review explores the molecular underpinnings of zeitgeber signals and critically assesses the translational barriers to implementing chrononutrition across species. By integrating insights from both preclinical and clinical research, we aim to refine the potential of circadian-based dietary interventions for metabolic disease prevention and personalized nutrition.

## 1. Introduction

The circadian clock is a fundamental regulator of nearly all biological processes, orchestrating molecular, physiological, and behavioral rhythms in alignment with the 24 h light–dark cycle. This endogenous timekeeping system ensures that key metabolic processes are optimally timed to anticipate and respond to environmental changes [1,2]. The central pacemaker, located in the suprachiasmatic nucleus (SCN) of the hypothalamus, synchronizes peripheral clocks in metabolically active tissues such as the liver, adipose tissue, and skeletal muscle [3]. While light serves as the primary zeitgeber for the central clock, feeding behavior plays a dominant role in entraining peripheral clocks, highlighting the intricate interplay between circadian biology and metabolism [4,5]. While our understanding of circadian biology has advanced considerably, the precise mechanisms through which meal timing, nutrient composition, and circadian rhythms shape metabolic health remain incompletely understood. Critical gaps remain in defining the molecular basis of meal timing-driven metabolic improvements and how this insight can be leveraged for clinical applications.

Recent advances in chrononutrition—the study of how meal timing influences circadian rhythms and metabolic health—have underscored the profound impact of circadian misalignment on metabolic health. Disruptions in the coordination between central and peripheral clocks, whether due to genetic, environmental, or behavioral factors, have been linked to metabolic diseases, such as obesity, insulin resistance, and cardiometabolic disorders [6,7]. Notably, alterations in meal timing, shift work, and irregular eating patterns can desynchronize metabolic pathways, exacerbating the risk of metabolic syndrome [8,9,10]. Increasing evidence supports the concept that synchronizing food intake with the body’s circadian rhythms, particularly through diet interventions such as time-restricted feeding (TRF), can improve metabolic health and lower the risk of chronic diseases. Crucially, the timing of meals, regardless of calorie intake, has been shown to impact metabolic function by promoting circadian alignment and improving metabolic resilience [11,12,13,14].

Despite these advances, several challenges hinder the translation of chrononutrition research into clinical practice. A significant obstacle lies in the reliance on murine models, as mice are nocturnal, feeding primarily during the dark phase, whereas humans are diurnal, with metabolic activity peaking during the day. This fundamental difference complicates the extrapolation of findings from rodent studies to humans, particularly when investigating dietary interventions, such as TRF. Additionally, many human studies face challenges related to variability in lifestyle factors, adherence to controlled feeding protocols, as well as chronotype—the individual’s timing preference for carrying out activities throughout the day and night—making direct comparisons between preclinical and clinical studies difficult [15,16]. Addressing these discrepancies is critical for refining the translational relevance of chrononutrition and developing personalized meal-timing strategies for metabolic disease prevention.

In this review, we explore the molecular and physiological basis of the circadian system and its impact on metabolic health. We discuss the emerging evidence on peripheral clocks in key metabolic tissues, the role of feeding behavior as a dominant zeitgeber, and the implications of meal timing for metabolic disease prevention. Furthermore, we highlight the challenges arising from murine and human studies, both at the anatomical and molecular levels. By integrating insights from molecular biology and nutrition science, this review aims to provide a comprehensive framework for leveraging circadian rhythms in dietary interventions to optimize metabolic health and disease management.

## 2. The Mammalian Circadian System

The molecular foundation of circadian rhythmicity resides in an autonomous molecular oscillator present in nearly all fully differentiated cells [2]. This highly conserved transcription–translation feedback loop operates on a 24 h cycle, wherein transcriptional activators induce the expression of their own repressor, thereby sustaining rhythmic gene expression. At the organismal level, circadian clocks are organized hierarchically, encompassing distinct classes of clocks with specialized roles that collectively orchestrate the mammalian circadian system, ensuring the precise synchronization of molecular rhythms with physiological and behavioral outputs [1,3,7].

Although the molecular oscillator maintains a phase approximating 24 h under free-running conditions, its period does not perfectly align with the external day. The system’s remarkable plasticity enables daily re-synchronization to environmental cues, or zeitgebers, with the light–dark cycle serving as the most consistent and universal signal across phylogenetic levels [17]. This dynamic interplay between intrinsic molecular mechanisms and external cues allows the circadian system to maintain alignment with the external environment, ensuring robust temporal regulation across biological scales.

### 2.1. The Circadian Central Clock and Its Regulatory Components

The mammalian circadian system is hierarchically organized, with the SCN functioning as the central “master clock” [1,3]. This master oscillator synchronizes molecular clocks in peripheral tissues, ensuring coordinated phase alignment throughout the body. Light acts as the dominant zeitgeber, maintaining the SCN’s ~24 h rhythm. Intrinsically, photosensitive retinal ganglion cells in the retina, which express melanopsin, detect environmental light and relay signals to the SCN via the retinohypothalamic tract (RHT) that facilitates adaptation to the external light–dark cycle.

The RHT plays a critical role in entraining the circadian system by transmitting photic signals directly to a subset of SCN neurons, particularly those located in the ventrolateral (core) region [18]. These neurons respond to glutamate and pituitary adenylate cyclase-activating polypeptide (PACAP), the primary neurotransmitters released by the RHT terminals. Upon light stimulation, these excitatory signals activate intracellular signaling cascades, including calcium influx and phosphorylation of CREB (cAMP response element-binding protein), which in turn modulates the expression of clock genes such as *Per1* and *Per2* [19,20]. This rapid gene induction enables the SCN to adjust its phase in response to changes in light exposure, forming the molecular basis of photic entrainment.

The timing and intensity of light input through the RHT are crucial, as they determine whether the clock is advanced or delayed [21,22]. For example, light exposure in the early subjective night typically delays the circadian phase, while light during the late night induces phase advances [23,24]. This bidirectional phase-shifting capacity allows the SCN to align internal rhythms with the external environment and maintain temporal homeostasis. Disruption in RHT signaling—whether due to retinal damage, altered light environments, or aging—can impair SCN entrainment and lead to downstream circadian misalignment in peripheral tissues [25,26,27,28]. By integrating environmental light cues through the RHT, the SCN is able to regulate the timing of behavioral and physiological processes, such as sleep–wake cycles, feeding, hormone secretion, and body temperature, thereby orchestrating systemic circadian harmony.

At the molecular level in the SCN, clock genes form the core of the transcriptional–translational feedback loop (TTFL) that operates the circadian oscillator [2]. The core loop is driven by the heterodimeric complex of CLOCK and BMAL1, which binds to E-box elements in the promoter regions of target genes, activating the transcription of core clock genes, such as Period (*Per1*, *Per2*, and *Per3*) and Cryptochrome (*Cry1*, and *Cry2*) [2]. The resulting PER and CRY proteins accumulate in the cytoplasm, undergo post-translational modifications (e.g., phosphorylation) and form complexes that translocate to the nucleus. Within the nucleus, these complexes inhibit CLOCK-BMAL1 activity, forming a negative feedback loop that reduces their own transcription. As PER and CRY proteins gradually degrade, repression is lifted, and a new cycle begins, completing the ~24 h oscillation. To add robustness to this system, an auxiliary loop regulates *Bmal1* expression. Nuclear receptors REV-ERBα/β and RORα/γ compete for binding to ROR response elements (ROREs) in the *Bmal1* promoter [1,2]. REV-ERBs act as transcriptional repressors, while RORs act as activators, creating a balance that fine tunes the oscillatory dynamics of *Bmal1* and, thus, the activity of the core loop. Beyond these core and auxiliary loops, PAR-bZIP transcription factors DBP and the repressor NFIL3 play a crucial role in an additional feedback mechanism called the accessory loop. Through binding to D-boxes, both DBP and NFIL3 integrate circadian control with downstream physiological processes (Figure 1). This multi-tiered regulation of central clock genes not only ensures precision and robustness of circadian rhythms but also allows dynamic adaptability to external cues, such as light and feeding, while coordinating a wide array of physiological processes.

### 2.2. The Peripheral Clocks

Beyond the master clock in the SCN, nearly all cells in the body host independent, self-sustaining clock regulatory systems as described in Figure 1 [1,2,3,7]. Through the modulation of core body temperature, autonomic nervous system signals, and endocrine signaling, including glucocorticoids and melatonin secretion, the SCN creates systemic rhythmic signals that reinforce the alignment of peripheral clocks [29,30,31]. Peripheral clocks, influenced by SCN-generated signals, regulate metabolic pathways and nutrient utilization, with oscillations in metabolites observed across tissues and blood plasma (Figure 2). Disruption of master and/or peripheral clock function abolishes the rhythmic patterns in activity and feeding behavior, resulting in a loss of metabolic rhythmicity and imbalance in energy expenditure, highlighting the essential interaction between central and peripheral clocks in maintaining metabolic equilibrium [30,32]. As tissue evolves to perform specialized functions, the importance of different environmental cues as zeitgebers adjust to match the specific demands of those functions, underscoring the tissue-specific roles of circadian rhythms in regulating metabolic and physiological functions, as exemplified by key peripheral systems such as adipose tissue, liver, skeletal muscle, and gut microbiota [1,3,7].

### 2.3. Adipose Tissue

Adipose tissue was initially recognized as an energy storage tissue, but it is currently known to have an endocrine function by secreting various factors, referred to as adipokines, and a thermogenic function [33,34,35]. The expression of clock genes, such as *Bmal1*, *Per1*, *Per2*, *Per3*, *Cry1*, and *Cry2,* and clock-controlled downstream genes, such as *Rev-erbα* and *Rev-erbβ*, show 24 h rhythmicity in inguinal white adipose tissue (iWAT), epididymal white adipose tissue (eWAT), and brown adipose tissue (BAT). Furthermore, the rhythmicity of the clock genes or adipokines in perigonadal adipose tissue is attenuated in obese mice [36,37,38]. Alteration of the adipocyte clock function leads to shifts in the timing of plasma polyunsaturated fatty acid levels, which subsequently influence the expression of neurotransmitters involved in appetite regulation within hypothalamic feeding centers [37]. These effects arise independently of changes in the rhythmic activity of circadian clock genes, such as *Bmal1*, within the hypothalamus, indicating that the adipocyte circadian clock directly influences hypothalamic feeding centers, without involvement from the local hypothalamic clocks [39]. Another study demonstrated that leptin expression in the adipose tissue is regulated by BMAL1/CLOCK-modulated binding of C/EBPα to the leptin promoter, resulting in the rhythmic transcription of leptin in adipose tissue and the circulating system, independently of food intake [40]. Leptin is an adipose tissue signature adipokine, acting as a strong inhibitor of appetite in the hypothalamus, and leptin resistance is commonly developed in obese subjects [41,42].

Modulation of clock genes has been shown to have a causal role in regulating adipose tissue function [37]. A study by Nam et al. showed that deletion of BMAL1 induced brown adipocyte differentiation [43]. They demonstrated that *aP2*-Cre *Bmal1*^flox/flox^ mice showed reduced obesity-induced whitening of BAT and lost more weight during the 24 h cold tolerance test compared to their littermate controls. However, despite this, *Bmal1*-deficient mice exhibited increased food intake and developed obesity when fed a chow diet [24]. It was also reported that depletion of BMAL1 from the adipose tissue using *Adipoq*-Cre mice model resulted in a significant increase in adipocyte size in eWAT without inducing adipose tissue inflammation [44]. These findings indicate that the function of clock genes varies depending on the adipose tissue types.

Chronic inflammation within white adipose tissue, marked by a shift toward production of pro-inflammatory adipokines, is central in the progression of obesity-induced systemic metabolic dysfunction [45,46]. *Clock* genes are known to regulate immune cell responses. REV-ERBα was demonstrated to repress *Ccl2* expression by directly binding the *Ccl2* promoter region [45]. CCL2 expression is known to be upregulated in obese mice, responsible for the recruitment of other immune cells [46,47]. In addition, myeloid cell-specific deletion of *Per1* and *Per2* mice showed an increase in body weight and visceral adipose tissue weight, as well as worsening adipose tissue inflammation and insulin resistance [48]. These findings indicate that clock genes play a role in preserving the homeostasis of adipose tissue.

### 2.4. Liver

The liver is a key organ in systemic metabolism, responsible for storing glycogen, synthesizing proteins, secreting hormones, and detoxification [49,50]. In mammals, the liver’s transcriptome and proteome exhibit rhythmic regulation driven by direct circadian control, along with cyclic modulation of nuclear transport and post-translational modifications [51,52]. As a key regulator of metabolic functions, the liver clock is influenced by dietary composition. For instance, an obesogenic diet significantly disrupts the rhythmic expression of thousands of liver genes [53,54]. Moreover, ketogenic diets amplify BMAL1 binding to chromatin and enhance the rhythmic activity of PPARα signaling in the liver [55]. On the other hand, acute alcohol drinks induce the rhythmic and expression level of the SREBP1 pathway, whereas chronic alcohol drinks suppress it [56,57,58]. These findings highlight how dietary factors and substance intake can distinctly modulate circadian regulatory pathways in the liver.

The disruption of specific clock genes in the liver has profound effects on the regulation of metabolic pathways. Liver-specific *Bmal1*-knockout resulted in arrhythmic expression of glucose regulatory genes and exacerbated glucose clearance [59,60]. Another study demonstrated that liver-specific *Bmal1*-knockout led to elevated plasma levels of low-density lipoprotein (LDL) and very low-density lipoprotein (VLDL) cholesterol, attributed to disruptions in the proprotein convertase subtilisin/kexin type 9 (PSCK9) and LDL receptor regulatory pathway [61]. Furthermore, deleting BMAL1 in the liver reduced the rhythmic expression of retinol-binding protein-4, which has been linked to insulin resistance, followed by enhanced systemic insulin sensitivity [62,63]. In response to fasting conditions, CRY1 and CRY2 inhibit cAMP-responsive element-binding protein (CREB) signaling in the liver by modulating G protein-coupled receptors and the stimulatory G-protein α-subunit [64,65]. Furthermore, the nuclear receptor REV-ERBα regulates the liver’s daily activation of SREBP-1C, which subsequently controls the expression of genes involved in hepatic cholesterol and lipid metabolism [66].

### 2.5. Skeletal Muscle

Skeletal muscle constitutes approximately 40% of body mass, playing a crucial role in locomotion, maintaining posture, and serving as a key metabolic organ involved in basal metabolism, glucose utilization, and fatty acid oxidation [67]. Given its significant contribution to systemic energy homeostasis, skeletal muscle has emerged as a promising therapeutic target for the prevention and treatment of metabolic diseases. A DNA microarray study by McCarthy et al. revealed 215 circadian genes in mouse skeletal muscle, with nearly 40% showing peak expression during the middle of the active (dark) phase [68]. Separately, four weeks of forced treadmill exercise during the inactive (light) phase shifted the circadian phase of bioluminescence rhythms in the skeletal muscle of PER2-Luciferase knock-in mice by three hours, while the SCN’s timing remained unchanged [69], indicating that exercise is a zeitgeber cue for skeletal muscle clocks that is independent of the SCN.

Several studies have tried to uncover the importance of the molecular clock for maintaining skeletal muscle homeostasis. *Bmal1*-null and *Clk/Clk* mutant mice showed dysfunctional mitochondria, reduced muscle force, and disrupted myofilament [70]. Additionally, *Bmal1*-null mice develop severe sarcopenia at 40 weeks of age [71]. However, it remains possible that the changes observed in the skeletal muscle of *Bmal1*-null and *Clk/Clk* mutant mice were indirectly caused by clock gene disruptions in other organs, as the skeletal muscle phenotype is strongly shaped by systemic factors, including hormonal signals and metabolic conditions. By using *Mlc1f*-Cre *Bmal1*^flox/flox^ mice system, Dyar et al. showed that skeletal muscle-specific KO of *Bmal1* mice gain more weight after ~30 weeks of age compared with their littermates, followed by the impairment of insulin-induced glucose uptake by muscle, without systemic glucose metabolic dysfunction [72]. To exclude the influence of *Bmal1* deletion on skeletal muscle development and postnatal maturation, Dyar et al. developed a tamoxifen-inducible muscle-specific *Bmal1* knockout (im-*Bmal1* KO; *HSA*-Cre-ER *Bmal1*^flox/flox^). They found no alteration in muscle weight and force between two littermates after 5 months (~20 weeks) of tamoxifen administration [72]. In a comparable im-*Bmal1* KO model, Schroder et al. observed decreased specific tension and elevated muscle fibrosis following 58 weeks of tamoxifen treatment [73]. The discrepancies between these two studies might be caused by the differences in experimental design, such as the age differences when tamoxifen was injected into the mice and the age of mice when they were analyzed.

### 2.6. Gut Microbiota

The gut microbiota, a diverse ecosystem of microorganisms, including bacteria, fungi, viruses, and archaea, functions as an integral part of the body, interacting closely with major systems, such as the immune, metabolic, and nervous systems. Disruptions in the balance of this microbial community can lead to significant health consequences, promoting the onset of chronic conditions, such as obesity, diabetes, metabolic syndromes, cancer, and autoimmune disorders [74,75,76]. A study by Thaiss et al. demonstrated that around 60% of the microbiota composition oscillates over a 24 h period [77]. Additionally, other studies revealed that, despite the absence of direct light exposure, gut microbes exhibit rhythmic fluctuations in both their population and functionality, driven by diurnal signals from the host [78,79,80]. For instance, *Bmal1*-null mice ablate the composition and abundance of bacteria in the feces [81]. Additionally, the whole-body knock-out of *Per1*/*Per2* alters the diurnal fluctuation of the gut microbiome, contributing to obesity and glucose intolerance [78]. Simultaneously, gut microbiota may contribute to the alignment of the host’s circadian clock by emitting various natural signaling molecules. Studies have uncovered that in mice lacking gut microbes, antibiotic-treated mice, or germ-free mice, alterations in circadian gene expression and metabolic pathways have been observed, suggesting a possible regulatory role of the gut microbiota on host circadian rhythms [78,79,80]. The gut microbiota influences the host’s circadian clock system through the activity of numerous bacterial metabolites, such as short-chain fatty acids, that have been identified as key players, displaying diurnal fluctuations in their levels [79,80,81,82].

## 3. Feeding Behavior as a Key Driver of Peripheral Rhythms Metabolic Health

Light–dark cycles serve as the primary zeitgeber for the master clock in the SCN, while feeding behavior is the most influential zeitgeber for peripheral clocks (Figure 3). In metabolically active tissue, such as the liver and adipose tissue, circadian rhythms control the expression of key genes, including nuclear receptors, affecting up to 20% of the transcriptome [51,83]. Studies have demonstrated that restricting food availability to the inactive light phase in mice shifted the phase of peripheral tissue clocks, such as those in the liver, pancreas, heart, skeletal muscle, adipose tissue, and kidney, without affecting the SCN [6,84]. This decoupling between central and peripheral clocks is evident under both normal light–dark cycles and constant darkness, underscoring the SCN’s insensitivity to feeding cues and suggesting that peripheral metabolism may not provide feedback regulation to the master clock [4,85]. However, this assumption is contradicted by findings indicating that the SCN core clock gene expression can be altered by prolonged hypocaloric TRF and sustained calorie restriction, whereas a single instance of hypocaloric TRF has no impact on the SCN phase [86]. These findings suggest that while meal timing alone is not a significant zeitgeber for the SCN, chronic caloric deprivation clearly acts as one.

The circadian clock in peripheral organs is significantly influenced by both the diet composition and the timing of food intake, extending its role beyond simply acting as a synchronizing mechanism [1,2,3]. Mice on a high-fat diet show increased daytime feeding, blunted clock gene oscillations, and altered metabolic gene expression, particularly in adipose tissue and the liver [87,88]. This includes impaired recruitment of BMAL1-CLOCK complexes, disrupted expression of genes, such as *Bmal1* and *Per2*, and altered transcription of metabolic regulators, such as *Pepck*. Interestingly, these effects only become apparent under ad libitum feeding conditions. In contrast, TRF has been shown to mitigate the negative outcomes associated with a high-fat diet, including obesity, hyperinsulinemia, and hepatic steatosis. This implies that the irregular feeding and fasting rhythms might play a pivotal role in the negative effects of a high-fat diet observed in animal models [10,13,14].

These findings highlight the critical importance of maintaining regular fasting–feeding, such as alternate-day fasting and time-restricted eating, which have shown promising improvements in metabolic parameters, including insulin sensitivity, lipid profiles, and body weight regulation. While significant progress has been made in elucidating the molecular mechanisms by which feeding behavior affects metabolic health, the majority of these insights have been derived from murine models. However, translational studies in humans remain limited, and notable interspecies differences in circadian organization and metabolic regulation underscore the need for caution when extrapolating findings. A deeper understanding of how feeding behavior interfaces with the human circadian system at the molecular level is critical for optimizing its clinical applications and maximizing its therapeutic potential.

## 4. Comparison of Chronometabolism Regulation in Mouse and Human Models

Considering the differences between mice and humans, as metabolic responses to fasting and feeding cycles vary significantly, a comparative analysis of circadian regulatory mechanisms in both species can help identify key divergences that shape metabolic adaptation and refine dietary interventions for human applications.

While both mice and humans share the fundamental framework of circadian regulation, key molecular differences affect how the metabolism responds to fasting and feeding cycles. In mice, peripheral clocks in metabolically active tissues, such as the liver and adipose tissue, exhibit strong entrainment to feeding cycles, with genes such as *Rev-erbα*, *Pparα*, and *Sirt1* showing pronounced oscillatory activity [89,90,91,92]. These genes regulate lipid metabolism, gluconeogenesis, and mitochondrial function in response to feeding schedules, reinforcing meal timing as a dominant metabolic cue. In contrast, human metabolic rhythms rely more on hormonal fluctuations—particularly cortisol, insulin, melatonin, and leptin—leading to a weaker direct coupling between feeding and circadian gene expression [84,93]. This hormonal dominance suggests that, while mice exhibit meal-driven metabolic regulation, humans rely more on endocrine signaling to maintain homeostasis (Figure 4).

The rhythmic expression of metabolic genes also differs significantly between species [1,94]. For example, hepatic genes, such as *Srebp1c* and *Fas*, which govern lipid synthesis, display strong circadian oscillations in mice but are more stable in humans, likely due to the overriding effects of fasting and postprandial metabolic states [94,95]. Similarly, in rodents, BMAL1 and CLOCK regulate thermogenesis via UCP1 in brown adipose tissue, a process that is much less pronounced in humans, where lipid metabolism is more influenced by hormonal control than intrinsic circadian oscillations [89,95].

Additionally, differences in gut microbiota-driven metabolic oscillations further complicate comparisons. In mice, the gut microbiome exhibits strong circadian fluctuations, dynamically shifting in composition and metabolic function based on feeding times [77,96]. In contrast, human gut microbiota shows more stable, diet-dependent variations with a weaker circadian-driven reorganization [97]. This suggests that dietary interventions targeting gut microbiota may require different strategies in humans than those observed in rodent models.

Taken together, these molecular discrepancies underscore the complexity of translating findings from murine chronobiology to human physiology. The stronger reliance on feeding cycles in mice contrasts with the multifaceted regulation of metabolism in humans, where endocrine rhythms and behavioral factors play a more dominant role.

### 4.1. Anatomical Differences in Mice and Humans in Relation to Chronobiology

Beyond the molecular difference, the anatomical structure of organisms also plays a fundamental role in regulating biological rhythms, including those governing feeding behavior, metabolism, and energy homeostasis [98,99,100,101]. The differences in organ structure and function between mice and humans lead to variations in how circadian rhythms influence metabolism. Such structures include the digestive system, central clock (SCN and pineal gland), and metabolic organs (liver and adipose tissue) [60,101,102,103,104]. These subsequently determine the way circadian signals regulate nutrient absorption, energy storage, and metabolic rhythms. While the core circadian clock mechanisms are conserved across species, the anatomical differences between mice and humans lead to species-specific metabolic adaptations to meal timing and circadian regulation [101,102,103].

One of the primary anatomical differences affecting circadian metabolic regulation is the structure of the intestines, which influences how nutrients are absorbed and processed in alignment with biological rhythms [105]. In mice, the small intestine is relatively longer, and the cecum is larger, facilitating enhanced microbial fermentation of dietary components. This structure enables stronger rhythmic control of microbial activity. Such features allow the gut microbiome composition to oscillate in a more pronounced manner over the circadian cycle [106]. As a result, microbial metabolites, such as short-chain fatty acids (SCFAs), follow a strong circadian rhythm in mice, impacting systemic metabolic pathways [107].

In contrast, humans have a shorter cecum and a different gut microbiome composition, leading to less pronounced circadian oscillations in microbial activity [106,107]. Instead of relying on fermentation for energy regulation, humans depend more on hepatic metabolic rhythms for glucose and lipid homeostasis [107,108]. At the molecular level, circadian regulation of nutrient absorption in the intestine is controlled by rhythmic expression of genes, such as the sodium–glucose transporter (*Sglt1*) and peptide transporter (*Pept1*), which exhibit species-specific variations in circadian patterns [109,110,111]. In mice, glucose absorption peaks during their active (night) phase, whereas in humans, it is most efficient during the early daytime hours, reflecting differences in feeding behavior and metabolic demands. While mice display sharper oscillations in metabolic gene expression based on feeding–fasting cycles, humans integrate a more complex set of cues, including light exposure and social behaviors, to regulate metabolism [112,113].

These anatomical differences provide a foundation for understanding how circadian regulation of metabolism varies between mice and humans. To further critically examine these molecular differences, this section explores the SCN and pineal gland, liver, and adipose tissue, highlighting the species-specific mechanisms that govern circadian metabolic regulation (Table 1).

#### 4.1.1. SCN and Pineal Gland: The Central Clock and Its Role in Metabolic Control

The role of the SCN in regulating metabolic processes differs between mice and humans, primarily due to differences in pineal gland function and melatonin secretion patterns. In mice, the SCN is more responsive to feeding–fasting cycles, meaning that peripheral clocks in the liver, adipose tissue, and gut are entrained primarily by meal timing rather than light–dark cycles [84]. This feeding-driven metabolic entrainment enables mice to rapidly adapt their metabolic state to shifts in food availability, a necessary adaptation for nocturnal survival strategies. In contrast, human SCN activity is more tightly coupled to light exposure, and while feeding can still influence peripheral clocks, circadian metabolic regulation in humans is primarily mediated through melatonin and hormonal signaling pathways [1,21,84].

A key molecular difference arises in the regulation of melatonin secretion. In humans, melatonin levels peak at night, inhibiting insulin secretion and promoting lipid oxidation over glucose metabolism [116,118,121]. This hormonal shift explains why late-night eating is associated with insulin resistance and metabolic dysregulation in humans. However, mice melatonin plays a minimal role in metabolic regulation, and feeding cues exert stronger control over clock gene oscillations in peripheral tissues. At the molecular level, *Bmal1* and *Clock* expressions in the SCN demonstrate stronger food-driven oscillations in mice, whereas in humans, these clock genes exhibit a more stable phase alignment with sleep–wake cycles [114,115,117,119,120].

These species-specific differences underscore the fundamental divergence in how circadian rhythms regulate metabolism, with mice relying more on direct feeding cues and humans integrating multiple external zeitgebers (time cues) to maintain metabolic homeostasis.

#### 4.1.2. Melatonin: A Bidirectional Signal Linking Light, Darkness, and Metabolism

Melatonin is a hormone primarily secreted by the pineal gland under the control of the SCN with a well-defined circadian rhythm [129,130]. Its levels begin to rise in the evening, peak during the night, and drop toward morning. This pattern is tightly entrained to the light–dark cycle via the RHT, which relays photic information from melanopsin-expressing retinal ganglion cells directly to the SCN [131,132] (Figure 5). In this way, melatonin serves as a key endocrine output of the central clock, signaling the onset of biological night to peripheral tissues. However, melatonin’s role is not unidirectional, rather, it also feeds back to the central oscillator [133]. Melatonin receptors (MT1 and MT2) are expressed within the SCN, and melatonin acts as a darkness signal, reinforcing the phase and amplitude of circadian rhythms [133]. In other words, melatonin is not only regulated by the SCN but also helps stabilize and refine SCN oscillations, particularly in response to environmental darkness. This feedback loop enhances circadian robustness and contributes to the internal synchronization of peripheral clocks [133,134,135].

While this dual role is well documented in humans, it is notably attenuated or absent in commonly used laboratory rodents, such as mice and rats, which produce little to no melatonin due to mutations or low pineal activity [136,137]. In humans, melatonin is a reliable marker of circadian phase and internal night [138]. In contrast, C57BL/6 mice, one of the most widely used murine strains, lack robust melatonin rhythms and, thus, do not provide an accurate model for melatonin-mediated circadian feedback [137,139,140]. As a result, rodent studies may overlook melatonin’s contribution to circadian entrainment and its downstream metabolic regulation. In humans, melatonin has been implicated in glucose metabolism, insulin sensitivity, and brown adipose tissue thermogenesis, with MT1/MT2 receptor signaling influencing lipid metabolism and inflammatory tone in peripheral tissues [129,141]. Clinical studies have shown that disrupted melatonin rhythms—such as through light exposure at night or shift work—are associated with increased risk of obesity, insulin resistance, and metabolic syndrome [142,143,144,145,146].

These differences emphasize the translational limitation of melatonin-deficient rodent models in chrononutrition and circadian research. Unlike in mice, where feeding rhythms dominate peripheral entrainment, in humans, melatonin provides an important systemic cue that integrates environmental light signals with internal metabolic regulation [134,147,148,149].

#### 4.1.3. Liver: Circadian Control of Glucose and Lipid Homeostasis

The liver is a key metabolic organ that governs glucose homeostasis, lipid metabolism, and amino acid turnover under circadian control. However, the degree of circadian metabolic entrainment in the liver varies between species, reflecting their distinct feeding patterns and hormonal regulatory mechanisms.

In mice, the liver exhibits strong circadian oscillations in metabolic gene expression, driven primarily by feeding–fasting cycles rather than hormonal cues [124]. This results in a robust rhythmic pattern of hepatic glucose production, lipid oxidation, and bile acid synthesis, with metabolic pathways peaking during the active (night) phase when mice are feeding. REV-ERBα and PPARα, which are the key transcription factors that regulate lipid metabolism, show strong feeding-induced oscillations in murine livers. In contrast, in humans, their rhythmic activity is more influenced by hormonal fluctuations, such as cortisol and insulin [122,123,125].

In human livers, metabolic regulation is more hormonally driven, with hepatic circadian rhythms responding primarily to endocrine signals rather than direct meal timing [111]. While meal timing does influence metabolism in humans, glucose production and lipid homeostasis are controlled by insulin sensitivity rhythms, cortisol fluctuations, and autonomic nervous system activity [150,151,152]. As a result, disruptions in sleep patterns and shift work have profound effects on human hepatic metabolism, leading to increased risks of non-alcoholic fatty liver disease (NAFLD), type 2 diabetes, and metabolic syndrome—a phenomenon less pronounced in mice due to their greater metabolic plasticity.

Molecular studies further highlight these species differences. In mice, hepatic *Per2* expression follows a strict feeding-related rhythm, peaking during the dark phase to regulate gluconeogenesis and lipid metabolism [153]. On the other hand, in humans, *PER2* oscillations are less sensitive to feeding but are more aligned with sleep–wake cycles and cortisol rhythms [84,134,154]. These findings suggest that, while circadian control of liver metabolism is conserved across species, the degree of feeding-driven versus hormonal regulation differs, influencing how meal timing affects metabolic homeostasis in each species.

#### 4.1.4. Adipose Tissue: Species-Specific Regulation of Energy Storage and Thermogenesis

Adipose tissue plays a vital role in energy storage, thermoregulation and endocrine signaling [34]. However, the regulation of adipose tissue function differs notably between mice and humans. In mice, adipose tissue exhibits strong circadian rhythmicity, whereas in humans, its function is predominantly governed by endocrine signals and homeostatic feedback [155].

Research indicates that, unlike in rodents, where Ucp1 expression exhibits strong circadian patterns, human Ucp1 is more consistently regulated by external factors, such as cold exposure and hormonal signaling, rather than the intrinsic circadian mechanism [127,128]. On the other hand, in humans, *Ucp1* expression is primarily regulated by external factors, such as cold exposure and thyroid hormones, with minimal modulation by the circadian rhythm [156,157,158,159]. In the context of lipid metabolism, lipolysis in mice follows a similar pattern, aligning with the feeding–fasting cycle and adrenergic signaling, marked by rhythmic activation of lipases such as HSL and ATGL during the active phase [160,161]. Meanwhile, in humans, lipolysis is predominantly driven by cortisol and insulin levels, peaking in the early morning and gradually decreasing with rising insulin levels, illustrating a hormone-dominant, less rhythmic pattern of fat mobilization [161,162].

Adipokines, such as leptin and adiponectin, display species-specific circadian dynamics. In mice, leptin levels peak during the fasting (light) phase under the regulation of clock genes, such as *Rev-erbα* and *Bmal1,* aligning appetite and energy use with feeding behavior [40,93]. In humans, leptin shows a less pronounced diurnal pattern, gradually rising in the evening in response to energy intake rather than circadian control [93]. Adiponectin also varies rhythmically in mice with energy status, while in humans, its secretion is relatively constant, reflecting nutritional and metabolic conditions. These differences emphasize the strong influence of the internal clock in mice and the greater role of systemic and environmental inputs in humans.

## 5. Bridging the Translational Gap: Limitations of Murine Models and Emerging Diurnal Alternatives

Although murine models have significantly advanced our understanding of the molecular mechanisms governing circadian rhythms, their nocturnal physiology presents significant challenges in translating findings to the diurnal human system. A more physiologically aligned option, diurnal animals, such as pigs (*Sus scrofa*) and degus (*Octodon degus*), have been considered to bridge this translational gap [163,164,165]. These species demonstrate circadian behavior and feeding patterns more closely aligned with humans, making them valuable translational models. The comparison of the different types of chronotype is summarized in Table 2.

Pigs are omnivorous, diurnal mammals that share numerous metabolic, gastrointestinal, and cardiovascular traits with humans [168,169,170]. Importantly, their activity–rest cycles and feeding windows are synchronized to the light phase, mirroring the circadian alignment found in humans [163,186]. At the molecular level, pigs demonstrate rhythmic expression of core clock genes in peripheral tissues, such as liver and adipose tissue, with patterns that closely resemble human circadian gene profiles [187,188,189]. Additionally, pigs exhibit postprandial hormonal dynamics (e.g., insulin, glucagon, GLP-1) that respond to meal timing in ways more analogous to humans than to mice [170,171]. Compared to nocturnal rodents, pigs offer a more relevant model for evaluating the impact of feeding time on metabolic homeostasis, particularly in TRF protocols where phase alignment with the light period is crucial. Their larger body size also facilitates repeated blood sampling and continuous metabolic monitoring, which are more limited in small rodents.

On the other hand, degus are diurnal rodents with robust circadian rhythms and an SCN architecture that closely resembles that of humans [165,166,167]. Molecular studies have shown that degus possess a functional circadian clock with light-responsive *Per1* expression in the SCN, and their peripheral tissues display circadian oscillations in metabolic gene expression that parallel those observed in humans [172,173,174]. Unlike mice, which often require reverse light–dark housing to match human rhythms, degus naturally align with human activity–rest patterns, making them particularly suitable for studying the timing of food intake, hormonal fluctuations, and behavioral responses during the active phase [175,176,177,178]. Furthermore, their small size and relatively short lifespan make them more accessible for longitudinal studies compared to larger diurnal mammals, such as pigs [165,179,180].

Nevertheless, while these models offer clear advantages, they also present notable practical and methodological limitations. Pigs require substantial space and care, making them costly and less practical for routine use in research [181]. In addition, they remain technically challenging for targeted genetic modifications due to restricted access to species-specific gene editing technologies [181,182,183,184]. While the pig genome has been sequenced, routine genetic engineering (e.g., tissue-specific knockouts) is not yet more widely available than in mice. Similarly, although degus are more practical than pigs, they lack the genetic toolkit and standardized protocols available as compared to the murine models [179,180,185]. Consequently, while diurnal models can serve as important translational intermediates, they may not fully replace the depth of mechanistic research possible in mice.

### Translating Pre-Clinical Findings to Human Trials

Despite the limitations of both nocturnal and diurnal animal models, murine studies remain central to preclinical chrononutrition research. These models have provided valuable mechanistic insights, particularly into how core clock genes regulate metabolic processes in response to feeding time. However, determining the translational relevance of these findings requires careful evaluation, as key physiological, behavioral, and environmental differences can significantly affect how these mechanisms manifest in humans (Table 3).

As one of the most widely studied chrononutrition strategies in animal models, TRF has consistently demonstrated strong metabolic benefits—even in the absence of caloric restriction [190,191,195,196]. Hatori et al. (2012) showed that TRF prevented obesity, hyperinsulinemia, and hepatic steatosis in high-fat-diet-fed mice, despite identical caloric intake to ad libitum controls [194]. Mechanistically, TRF in mice enhances BMAL1 and CLOCK binding to metabolic gene promoters, restores oscillatory expression of *Rev-erbα* and *Pparα,* and improves mitochondrial function in liver and adipose tissue [6,196]. These results suggest direct entrainment of peripheral clocks by feeding time, driving rhythmic metabolic gene expression independently of calorie intake.

In contrast, clinical trials in humans have yielded more modest, and often inconsistent, outcomes [192,197,198]. For instance, Sutton et al. (2018) reported that early TRF (eating from 8 a.m. to 2 p.m.) improved insulin sensitivity, blood pressure, and oxidative stress markers in prediabetic men—even without weight loss [14]. Jamshed et al. (2019) also observed improvements in 24 h glucose profiles and circadian hormone markers following early TRF [13]. However, these effects are less robust than those seen in mice and are not consistently replicated across studies [6,13,14,191,192,193,195,196,197,198].

These discrepancies underscore a fundamental translation gap. Unlike mice housed under tightly controlled conditions—with uniform light/dark cycles, diet, and genetics—humans are exposed to highly variable environments that interact with circadian and metabolic systems. Human metabolism is further influenced by complex behavioral and hormonal dynamics, including sleep quality, stress, melatonin–cortisol rhythms, and social eating patterns, which are difficult to model in animals [93,150,199]. Moreover, most human trials rely on indirect metabolic outcomes (e.g., glucose, lipid profiles) without assessing circadian gene expression directly [13,14,191,192,197], making it challenging to verify whether metabolic improvements are truly circadian-driven or behaviorally mediated.

Ultimately, while murine studies clarify the molecular basis of feeding–fasting rhythms, they oversimplify the context in which human metabolic health unfolds. Bridging this translational divide will require integrating mechanistic findings with human-specific variables through tools such as wearable circadian biomonitoring, tissue-specific metabolic profiling, and behavioral chronotype. Without these, the promise of chrononutrition risks being lost in the complexity of real-world application.

## 6. Human-Centric Limitation in Chrononutritional Research

In humans, the circadian timing of food intake has emerged as a groundbreaking approach for metabolic health [1,2,7,9]. This approach highlights the importance of aligning eating patterns with the body’s internal biological clock to optimize metabolic outcomes. Despite its promising implication, the advancement of research in this area has encountered several challenges, including methodological limitations and variability in human application [11,13,15]. Addressing these obstacles is essential to fully elucidate the clinical relevance and mechanistic underpinnings of TRF and other chrononutritional strategies (Table 4).

### 6.1. Inconsistent Methodologies

Current chrononutrition studies, mostly conducted in controlled settings, struggle with limitations in self-reported food intake methods, such as 24 h dietary recalls (24HRs), which suffer from memory bias, portion size estimation issues, and interviewer variability. Additionally, the lack of standardized tools for assessing behaviors leads to inconsistent findings, limiting research reliability and broader applicability [200,201,202]. Without universal methods, cultural and population-specific differences limit the generalizability of findings, while clinical applications and integration with emerging technologies remain constrained. This leaves chrononutrition as a field that remains fragmented, with limited impact on global health policies and practical interventions. Developing adaptable, validated tools is crucial to ensure their growth and relevance.

### 6.2. Individual Differences

Another key challenge in chrononutrition research is accounting for individual chronotype differences, which profoundly influence their physiological processes, including metabolic activity [213,214,215,216]. Studies have demonstrated that early chronotypes tend to have better glucose tolerance in the morning, while late chronotypes often show delayed peaks in metabolic efficiency [150,203,204]. However, current research struggles to capture these variations comprehensively, as many studies apply uniform meal timing protocols without considering individual circadian preferences. This one-size-fits-all approach introduces significant variability in outcomes and limits the ability to draw generalized conclusions. Additionally, genetic and epigenetic factors, such as variations in circadian clock genes (*Clock*, *Bmal1*, *Per*, and *Cry*), influence individual responses to meal timing, but research in this area faces challenges due to gene–environmental interactions and limited population diversity. To improve chrononutrition interventions, integrating personalized approaches that consider chronotype, genetic background, and lifestyle factors is crucial for optimizing metabolic health outcomes.

### 6.3. Long-Term Feasibility

Short-term chrononutrition studies show promising results, but their long-term feasibility is limited due to rigid, clock-based meal definitions that overlook cultural diversity and individual circadian rhythms [217,218,219]. This Western-centric approach reduces the applicability of findings, especially for populations with varied eating patterns, such as late-night dining in Mediterranean regions or dispersed meals in Asian cultures [128,129]. Practical constraints, including work schedules and social obligations, further challenge strict meal timing adherence, making flexible, personalized interventions essential for sustainable dietary strategies [220,221,222,223]. Additionally, while circadian realignment strategies, such as light exposure and melatonin supplementation, show potential, their success depends on adherence and addressing broader societal factors, such as shift work and urbanization, to ensure lasting health benefits [15,16,31,146,224,225].

### 6.4. Socio-Ecological Factor

While methodological inconsistencies, chronotype variability, and long-term feasibility have been widely recognized in chrononutrition research, these limitations often stem from a layer. Such a layer includes the socio-ecological context in which dietary behaviors occur [213,226,227,228,229]. Chrononutrition interventions do not operate in a vacuum; rather, they are shaped by a multilayered network of influences, ranging from individual traits to societal norms. At the individual level, metabolic responses to meal timing are modulated not only by chronotype but also by sleep quality, stress, and occupational demands, all of which are unequally distributed across populations [200,230,231,232,233,234]. At the interpersonal and community levels, cultural meal customs (e.g., evening feasting in Mediterranean cultures or late-night social eating in East Asia), religious fasting practices, and shared family eating routines may conflict with rigid TRF protocols [205,206,207]. At the societal level, structural determinants, such as socioeconomic status (SES), urbanization, and food insecurity, exert profound influence over individuals’ ability to adhere to time-based eating protocols [226,227,228]. Those in lower SES brackets are often subject to irregular work hours, limited food access, and financial constraints that favor calorie-dense, nutrient-poor foods [208,209,210,211]. These conditions not only compromise dietary quality but also lead to erratic meal timing, which directly conflicts with the structured patterns promoted in time-restricted feeding (TRF) interventions. In particular, food insecurity can result in unpredictable eating behaviors driven by availability rather than biological timing [235,236,237]. Thus, undermining circadian alignment and introducing stress-related metabolic disruptions. Likewise, urbanization and shift work promote 24 h lifestyles, increased exposure to artificial light at night, and inconsistent sleep and meal schedules [146,212]. These realities disproportionately affect marginalized populations and are rarely accounted for in clinical study designs, despite their significant impact on adherence and metabolic outcomes [238,239,240,241].

Adding another layer of complexity is the prevalence of Western-centric assumptions in chrononutrition research. Many current protocols are based on ideals of early dinner, consolidated daytime meals, and structured work–rest patterns [238,242,243]. These norms do not reflect the lived experiences of diverse global populations. In regions such as the Mediterranean, Middle East, and parts of Asia, culturally embedded practices often include late-evening meals, irregular eating occasions, or religious fasting patterns [207,244,245,246,247]. Applying rigid, one-size-fits-all TRF protocols in these contexts risks both poor adherence and limited clinical relevance. Moreover, few studies stratify outcomes by cultural background, SES, or occupational demands, making it difficult to generalize findings across populations [248,249,250]. Without explicit integration of behavioral, economic, and cultural contexts, even well-controlled interventions may fail to produce sustainable or equitable outcomes. To truly advance the field, chrononutrition research must adopt socio-ecologically informed frameworks that reflect the diversity of real-world lived experiences. This includes designing adaptable, culturally sensitive, and socioeconomically feasible strategies that can accommodate varying chronotypes, lifestyles, and resource availability. Only by addressing these contextual complexities can chrononutrition fulfill its promise as a globally relevant approach to metabolic health.

## 7. Future Direction of Chrononutrition Research for Obesity Management

Building upon the challenges that were discussed above, personalized dietary plans could provide solutions tailored to the individual’s needs. Such plans hold a prospective benefit that might be applicable in the management of obesity-related conditions [251,252,253,254]. Personalized dietary plans leverage the understanding that metabolic responses to meal timing are not uniform across individuals. Chronotypes, genetic predispositions, and lifestyle factors influence how and when the body processes nutrients, making a one-size-fits-all approach ineffective [214,253,255]. Accurate assessment of chronotype, genetic factors, and metabolic profiles is essential but resource intensive, requiring tools such as genetic testing and detailed dietary tracking. Furthermore, adherence to personalized plans depends on behavioral factors, such as motivation and the ability to navigate social or occupational constraints. By considering individual variability, these plans can reduce the risk of metabolic dysregulation associated with misaligned meal timing, offering a practical pathway to long-term obesity management [256,257]. Future research and technological advancements are critical to refining these approaches, making personalized chrononutrition interventions accessible and effective for diverse populations.

TRF has emerged as a powerful tool in chrononutrition by focusing on limiting food intake to specific time windows to align with the body’s circadian rhythms [5,6,13,14,196]. However, its effectiveness depends not just on fasting duration but also on nutrient timing within feeding windows. Optimizing macronutrient intake—such as consuming carbohydrates earlier when insulin sensitivity is highest and prioritizing protein during active periods—can enhance metabolic outcomes, yet many studies overlook this aspect [195,258,259]. Additionally, technology-driven solutions, including wearable devices and smartphone apps, can provide real-time tracking, personalized meal reminders, and dietary analysis, making chrononutrition strategies more accessible and actionable, though clinical validation remains essential for their widespread implementation [11,258,260,261,262,263]. As chrononutrition continues to gain recognition as a critical field in metabolic health research, its future directions must address gaps and expand into areas that hold transformative potential. One such area is the interplay between nutrient timing and gut microbiota. Recent findings highlight that the gut microbiota operates in circadian cycles, with its composition and activity fluctuating throughout the day [77,82,83,264]. Studies have demonstrated that TRF can restore diurnal microbial oscillations, reduce systemic inflammation, and enhance glucose metabolism [265,266,267]. However, the complexity of gut microbiota and the individual variability in its composition necessitate further research into how specific macronutrient timing, such as fiber or probiotic intake, modulates microbial activity to achieve sustained metabolic benefits. Furthermore, chrononutrition must address the unique needs of specific populations, including women, children, and the elderly, as these groups exhibit distinct circadian characteristics that influence metabolic outcomes [268,269,270]. Another promising direction lies in integrating chronopharmacology with chrononutrition, particularly in the timing of nutritional supplements to enhance their efficacy [271,272,273,274]. For instance, omega-3 fatty acids, known for their anti-inflammatory properties, could be timed to counteract periods of heightened inflammation [275,276]. These strategies not only enhance the therapeutic potential of supplements but also offer a targeted approach to managing metabolic dysregulation. To advance chrononutrition, these future directions must integrate multidisciplinary approaches that combine advancements in molecular biology, technology, and behavioral science. The field’s potential to revolutionize metabolic health and obesity management lies in its ability to deliver personalized, culturally sensitive, and biologically aligned interventions. By addressing the dynamic interplay between meal timing, individual variability, and population-specific needs, chrononutrition research can move beyond its current limitations to provide sustainable, impactful solutions for global health challenges. This future vision underscores the importance of continued innovation and collaboration in transforming chrononutrition from a promising concept into a cornerstone of preventive and therapeutic medicine.

## 8. Conclusions

Chrononutrition has emerged as a promising framework for optimizing metabolic health and managing obesity through the strategic alignment of food intake with the body’s circadian rhythms. At the core of this field lies a complex interplay between the central clock in the SCN, peripheral clocks in metabolic tissue, and external zeitgebers. This review outlines how TTFL orchestrates circadian rhythmicity at the molecular level. We further explored how differences in anatomical structures and signaling pathways influence circadian regulation, particularly the role of feeding and systemic mediators such as melatonin and glucocorticoids in synchronizing peripheral clocks. While mechanistic insights from rodent models have laid the groundwork for understanding circadian–metabolic interaction, translational challenges remain substantial. Key differences between nocturnal rodents and diurnal humans limit the applicability of current pre-clinical findings. Although alternative models, such as pigs and degus, offer some advantages, they have not yet fully resolved these translational gaps. Human studies are further complicated by confounding variables, such as individual chronotype differences, cultural dietary habits, socioeconomic factors, and variability in study design. Chrononutrition is not a one-size-fits-all strategy but rather a multifactorial paradigm that requires the integration of biology, behavior, and environment. Interdisciplinary efforts across molecular chronobiology, nutrition science, behavioral medicine, and public health are essential for unlocking its full therapeutic potential. With careful refinement, chrononutrition may become a foundational pillar in the prevention and management of metabolic disorders.

## Figures and Tables

**Figure 1 ijms-26-05116-f001:**
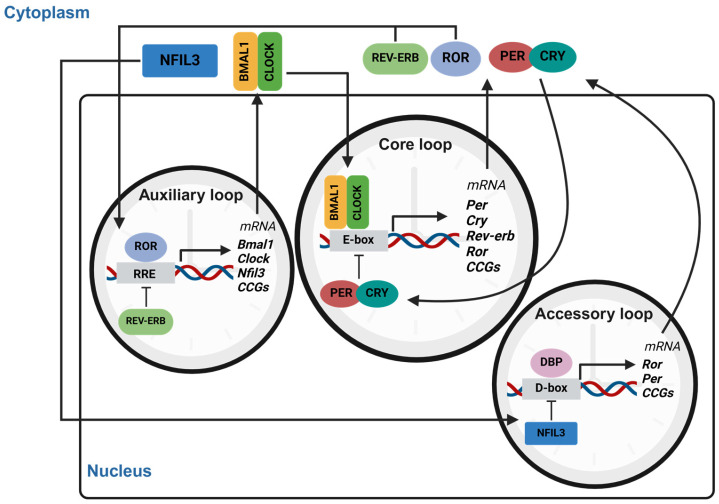
Molecular mechanism of circadian clock and its TTFL. The core loop is driven by BMAL1:CLOCK binding, which activates *Per*, *Cry*, *Rev-erb* and *Ror* transcription via E-box binding. PER and CRY proteins form complexes that inhibit BMAL1:CLOCK, closing the feedback loop. The auxiliary loop, involving the nuclear receptor ROR and REV-ERB protein, regulates *Bmal1* through RRE sites, while the accessory loop, including DBP and NFIL3 protein, modulates clock-controlled genes via D-box elements. These interconnected loops maintain circadian rhythmicity in gene expression. This image was created with BioRender (https://BioRender.com/9gga913).

**Figure 2 ijms-26-05116-f002:**
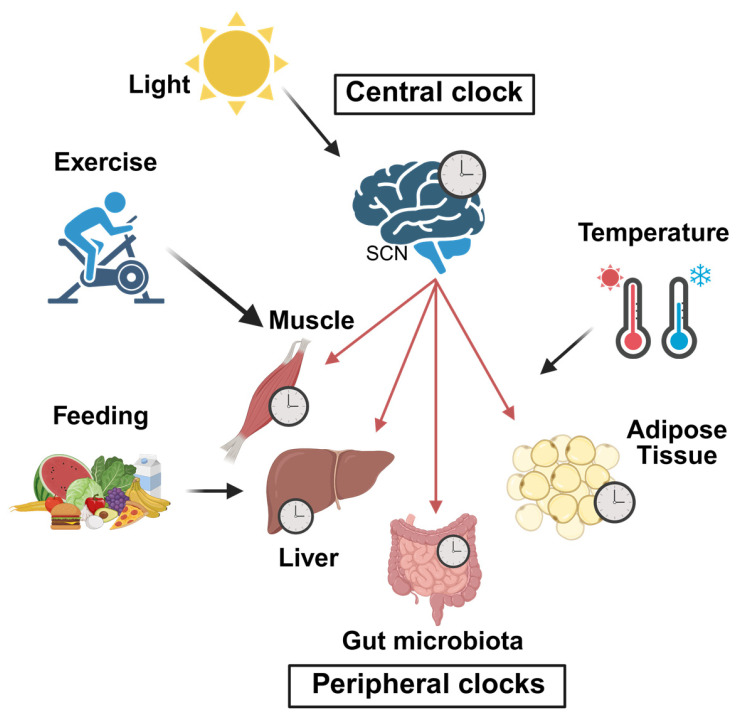
Regulation of central and peripheral clocks by environmental and physiological cues (black arrow). The central clock located in SCN is primarily entrained by light and coordinates systemic rhythmic signal to align peripheral clocks (red arrow) in metabolic tissue, such as liver, muscle, adipose tissue, and gut. Peripheral clocks can also respond directly to environmental cues, such as feeding, exercise and temperature. Such a process enables tissue-specific regulation of circadian rhythms in metabolism and energy homeostasis. This image was created with (https://BioRender.com/msvpxk9).

**Figure 3 ijms-26-05116-f003:**
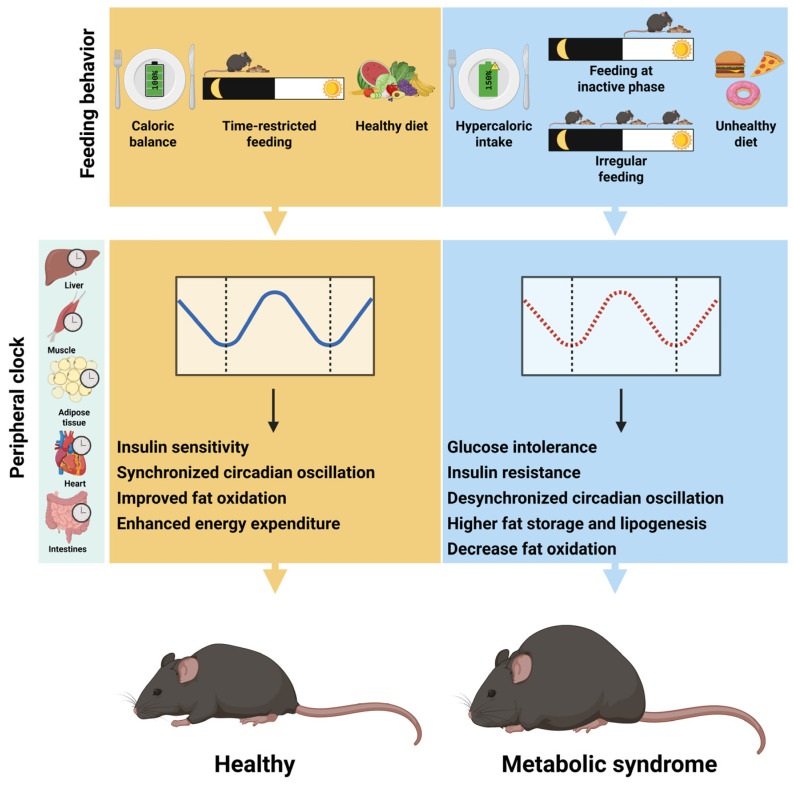
Impact of feeding behavior on peripheral clocks and metabolic health. This image illustrates how different feeding behaviors, such as caloric load, timing and dietary composition (top panel), can affect the peripheral clock gene oscillation (middle panel), thus affecting systemic metabolic homeostasis. This image was created with BioRender (https://BioRender.com/2fqad01).

**Figure 4 ijms-26-05116-f004:**
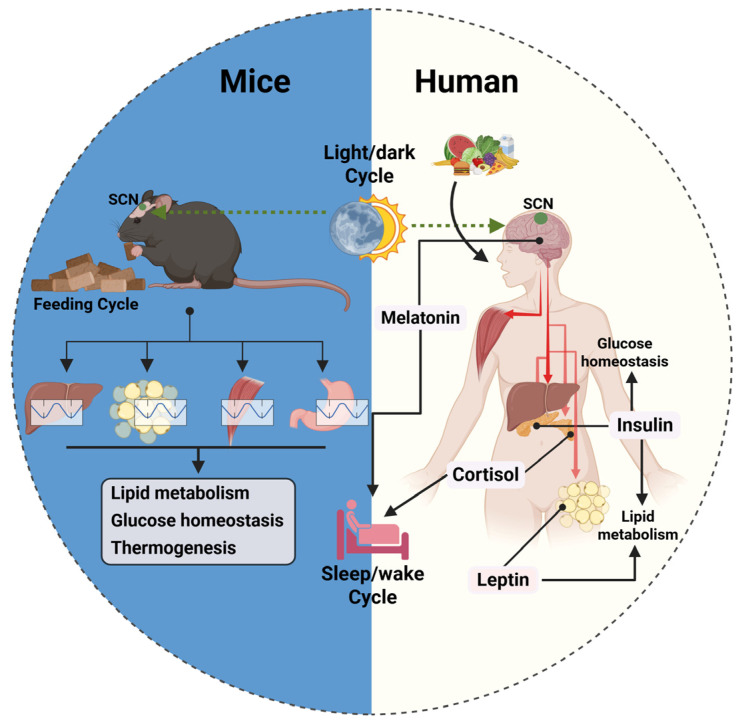
Distinct circadian regulation of metabolism in nocturnal rodents versus diurnal humans. In nocturnal rodents (**left**), feeding cycles serve as the dominant zeitgeber for peripheral clocks in metabolic tissues, such as liver, adipose, muscle, and gut. The alignment between feeding time and clock gene expression drives rhythmic metabolic processes. In contrast, human metabolism (**right**) is regulated by a more complex integration of behavioral patterns and hormonal rhythms orchestrated by the central clock in SCN. The SCN receives light input and modulates systemic signals, including melatonin, cortisol, and insulin, which in turn regulate peripheral clocks and lipid metabolism. These differences highlight the translational challenge of applying findings from murine models to human physiology. This image was created with BioRender (https://BioRender.com/b2vk839).

**Figure 5 ijms-26-05116-f005:**
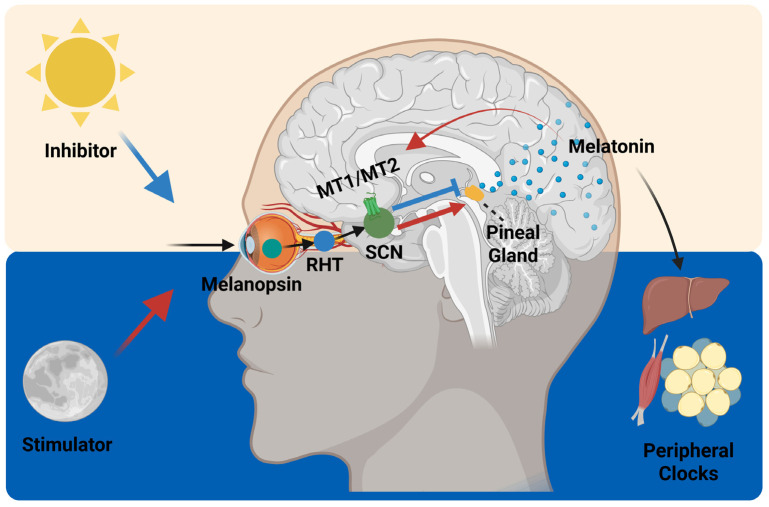
Melatonin-mediated regulation of central and peripheral clocks. Light signals received by melanopsin-expressing retinal cells are transmitted to the SCN via RHT. In darkness, the SCN activates melatonin secretion from the pineal gland, which synchronizes the peripheral clock in metabolic tissue. Light inhibits this pathway and aligns the circadian rhythm with the external environment. This image was created with BioRender (https://BioRender.com/o7ryg3a).

**Table 1 ijms-26-05116-t001:** Chronometabolic regulation comparison in mice and humans.

Chronometabolic Regulation	Mice (Nocturnal)	Humans (Diurnal)
SCN & Pineal gland regulation [1,21,84]	SCN entrains peripheral clocks primarily via feeding–fasting cycles.	SCN entrains peripheral clocks primarily through light exposure and hormonal signals.
Melatonin influence on metabolism [114,115,116,117,118,119,120,121]	Minimal melatonin influence: feeding cues exert stronger control over clock genes	Melatonin peaks at night, suppressing insulin and promoting lipid oxidation.
Circadian regulation of liver metabolism [122,123,124,125]	Strong circadian oscillations in hepatic gene expression, driven by meal timing.	Regulation is predominantly hormonal (cortisol, insulin, glucagon) rather than meal timing.
Regulation of thermogenesis & lipolysis [126,127,128]	Thermogenesis process is strongly regulated by BMAL1 & CLOCK expression. The lipolysis follows a strict circadian cycle (peaking at night)	Thermogenesis & lipogenesis is less circadian driven; influenced more by hormonal fluctuations
Circadian regulation of adipokines [40,90,93]	Leptin and adiponectin secretion follow a strong circadian rhythm, peaking during fasting.	Leptin secretion follows a gradual diurnal rhythm, peaking in the evening, influenced by energy balance.

**Table 2 ijms-26-05116-t002:** Comparative characteristics of diurnal and nocturnal animal model.

Traits	Pig (*Sus scrofa*)	Degu (*Octodon degus*)	Mouse (*Mus musculus*)
Chronotype [163,164,165,166,167]	Diurnal	Diurnal	Nocturnal
Metabolic similarity to humans [168,169,170]	High—similar gastrointestinal, cardiovascular, and metabolic traits	Moderate—robust circadian behavior	Low—opposite feeding/activity cycle
Peripheral clock gene expression [170,171,172,173,174]	Rhythmic expression similar to human in liver/adipose tissue	Functional *Per1* expression in SCN, peripheral oscillations	Highly studied; strong rhythmicity but reversed phase
Postprandial hormonal dynamics [170,171,175,176,177,178]	Responsive insulin, GLP-1, glucagon dynamics	Circadian-aligned feeding and hormonal patterns	Meal timing dominant; less human-like hormonal profiles
Ease of handling [165,179,180]	Difficult—large size, high cost	Easier—small, size, short lifespan	Very easy, widely used, low cost
Genetic manipulation tools [179,180,181,182,183,184,185]	Limited availability of gene editing tools	Lacks standardized genetic tools	Well-developed gene editing (e.g., CRISPR, Knockouts)
Practical limitation [165,170,180,181]	High housing cost, limited scalability	Limited protocol, less well characterized	Circadian mismatch with human

**Table 3 ijms-26-05116-t003:** Comparison of TRF outcomes in pre-clinical study and clinical trials.

Characteristic	Mice (Pre-Clinical Studies)	Humans (Clinical Trials)
Feeding pattern [190,191]	TRF aligned with active (dark) phase	TRF aligned with active (light) phase
Metabolic effect [192,193,194,195]	Strong; ↓ obesity, ↓ insulin resistance, ↓ liver fat	Mild to moderate: ↑ insulin sensitivity, ↓ BP
Mechanism assessed [6,196]	Yes—gene expression (*Bmal1*, *Clock*, *Rev-erbα*, *Ppar*α)	Rarely assessed directly
Environmental control [13,192,197]	Highly controlled (light, diet, genetics)	Variable (light exposure, chronotype, stress)
Clock gene oscillation [13,192,197]	Frequently reported	Rarely reported
Limiting factors [13,190,192,195,196,197]	Nocturnal model; limited translation	Poor adherence, high variability

**Table 4 ijms-26-05116-t004:** The limitations in human studies and their influence on TRF strategy.

Barrier Category	Description	Impact on TRF Adherence
Methodological Limitation [200,201,202]	Inconsistent dietary assessment tools and lack of standardized protocols reduce comparability and reliability.	High—unreliable intake data and measurement bias undermine study validity.
Chronotype Variability [150,203,204]	Differences in individual circadian preferences affect metabolic responses but are rarely accounted for.	High—one-size-fits-all protocols fail to address individual metabolic timing.
Cultural & Behavioral Practices [205,206,207]	Traditional meal customs, religious fasting, and family routines may conflict with rigid TRF schedules.	Moderate to high—sociocultural norms often override protocol adherence.
Socioeconomic Constraints [208,209,210,211]	Low SES limits food access, meal regularity, and flexibility to follow structured interventions.	Very high—financial, environmental, and access barriers prevent participation.
Work Schedules & Urbanization [15,16,31,146,196,212]	Shift work, artificial light, and 24 h lifestyles disrupt circadian alignment and eating patterns.	High—inconsistent daily structure interferes with stable meal timing.
Western-Centric Bias [205,206,207]	Early-meal TRF protocols are based on Western routines, limiting global applicability and equity.	High—reduces inclusivity and generalizability of research findings.

## Data Availability

Not applicable.

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
