# Peer review of "Chrononutrition: Potential, Challenges, and Application in Managing Obesity"

_ijms, 2025, doi:10.3390/ijms26115116_

Round 1

Reviewer 1 Report

Comments and Suggestions for Authors

This manuscript offers a comprehensive review of chrononutrition and its potential role in obesity management, integrating molecular mechanisms, interspecies differences, and translational challenges. The review makes a valuable contribution to the field of chrononutrition and its applications in metabolic health. While the topic is of significant interest to researchers, certain aspects could be further strengthened before publication.

1. The authors provide an extensive discussion of murine studies (e.g., BMAL1/Clock regulation in adipose tissue) but do not sufficiently address the limitations of translating these findings to humans. A dedicated section comparing human clinical trials with murine data would enhance the review’s translational relevance.

2. Some mechanistic claims remain speculative. For example, the assertion that "gut microbiota oscillations directly influence hypothalamic feeding centers" relies on indirect correlations (e.g., germ-free mouse studies) rather than causal evidence. Clarifying such statements (e.g., using "may influence" or "could contribute") or citing direct experimental evidence would strengthen the discussion.

3. Although the review acknowledges interindividual variability (e.g., chronotypes), it does not critically assess how confounding factors—such as socioeconomic status, cultural dietary practices, or sleep quality—may influence the outcomes of chrononutrition interventions. Addressing these aspects would provide a more nuanced perspective.

Reviewer 2 Report

Comments and Suggestions for Authors

Comments to the Author

The article "Chrononutrition: Potential, Challenges, and Application in Managing Obesity" addresses a significant topic in human medicine. The circadian clock regulates almost every bodily function, coordinating metabolic activities with environmental signals including light and food consumption. A vital connection between circadian biology and metabolism is shown by this interaction. When these mechanisms are disturbed, whether by shift work, erratic eating habits, or a misaligned lifestyle, metabolic illnesses like obesity, insulin resistance, and cardiometabolic diseases have been closely associated. Interindividual diversity in chronotype, behavioral habits, and nutritional compliance further complicates human investigations.

Addressing observations requiring clarification would improve the manuscript.

Consideration should be given to adding more species, or to specifying in the abstract that only a murine and a human model are involved.

Major Concerns:

Given the amount of information contained in this article, it would be useful for more figures to be added (chapters 3, 4, and 5). This would facilitate the understanding of the text.

To have many models, particularly diurnal ones, a comparison with the pig could be made.

Minor Issue:

In terms of presentation, the addition of more Tables should be considered.

In Figure 1 the suprachiasmatic nuclei (SCN) are not visible. Furthermore, the role of the retinohypothalamic tract (RHT) should be included, through which light information is sent directly from the retina to a subset of SCN neurons. The role of melatonin in this circadian rhythm should also be addressed. This pineal hormone, known as melatonin, is found to be most abundant in the blood at night and least prevalent during the day. Its secretion is governed by a rhythm-generating process   within the SCN, which is regulated by light. Not only is melatonin's regulation influenced by the circadian oscillator, but a feedback signal for darkness is also provided to the oscillator by melatonin itself. Consequently, the nocturnal mechanisms should be incorporated into this Figure.

A more specific and comprehensive conclusion should be provided. Such a response is warranted by the information in the manuscript.

Reviewer 3 Report

Comments and Suggestions for Authors

The manuscript entitled “Chrononutrition: Potential, Challenges, and Application in Managing Obesity” explores the molecular basis of zeitgeber signals, translational barriers regarding chrononutrition in humans and mice. The present work is well structured, written in a way that is very easy for any reader in the scientific field to follow and understand. In addition, the topic is innovative and interesting, in my opinion the paper can be accepted for publication after a minor revision.

Specific comments

Line 25 “across species” it is not appropriate to talk about different species, as this review refers to humans primarily and mice. This terminology seems to refer to other animal species as well.

Line 74-75 “both at the anatomical and molecular level”

Line 87-93 a recent bibliographical source should be included to support the section such as:

DOI: 10.1080/07420528.2024.2360723

Line 131-135 these information should be supported by references

Line 178 Please delete brackets in reference

Figure 1 the form the Per, Cry, Rev-erb and Ror transcription or ROR and REV-ERB should always be the same through the text

4.1.1; 4.1.2; 4.1.3 The subchapters should like the others be given in italics, please revise the text

Line 463 please delete “a”

Round 2

Reviewer 2 Report

Comments and Suggestions for Authors

Accept in present form.